# Write More at Once: Stylized Chinese Handwriting Generation via Two-stage Diffusion

## Abstract

Handwritten data generation is an intriguing research area with broad applications in human interaction with digital documents. In Chinese handwritten text generation, practical applications necessitate the ability to produce sentence-level handwritten data to convey complex information effectively. However, existing methods mainly focus on generating single-font outputs. To tackle this challenge, we model handwritten text generation as a *style transfer problem*, aiming to convert a standard text line template into a target handwriting style. Recognizing the highly structured nature of handwritten data, we view complex text lines as compositions of individual characters and their positions. We propose a two-stage text line generation method based on generative diffusion model. In the first stage, character positions are generated using a Character-Position-Diffusion (CharPos-Diff), which, combined with standard character templates from a digital library, creates text line-level templates. In the second stage, a font style transfer diffusion model (Imitating-Diff) generates handwritten text lines directly from these templates. Our extensive experiments show that our method effectively mimics handwriting styles, generates structurally accurate text lines, and facilitates the simultaneous generation of paragraph-level handwritten text.

## 1 Introduction

Handwritten data represents a significant intersection between the fields of Natural Language Processing and Computer Vision, as it encompasses both linguistic information and the visual characteristics of images. Unlike standardized printed text, handwritten data is highly personalized, reflecting each person's unique writing habits and styles, and is capable of conveying rich emotional information. The task of generating handwritten data aims to simulate the properties of authentic handwritten text through algorithms, enabling the automatic production of personalized handwritten content (Kang et al., 2021; Xu et al., 2022). This task must ensure that the generated characters are structurally correct and legible, while also visually preserving the natural style and diversity of handwriting. This technology has demonstrated significant potential across various applications, such as personalized digital signatures, handwriting-simulated email generation, personalized document displays, and even the reproduction and restoration of historical texts (Guan et al., 2024).

Among the multitude of languages, Chinese characters present considerable complexity due to their large quantity and intricate forms (Xu et al., 2009; Lin et al., 2015; Lian et al., 2018; Radford et al., 2016; Zhu et al., 2017), thereby posing additional challenges for the generation task. Early approaches could only generate fonts seen in the training set, while recent methodologies typically utilize few-shot learning techniques to adaptively mimic calligraphy style based on several provided style reference samples. However, these methods have *primarily focused on generating individual Chinese characters*, neglecting the need for generating text at the line or even paragraph level in real-world applications. Furthermore, due to the variability in individual writing habits, merely generating single characters and concatenating them in a model-free manner can lead to rigid and unnatural layouts, hindering practical applications.

In this paper, we model the handwritten data generation task as a style transfer problem that converts a given standard character template into a target handwritten style. In the first stage, we generate high-quality content templates by modeling the positional relationships of individual characters within a text line, in conjunction with a standardized character database. In the second stage, we employ image-to-image translation techniques to transform the template into stylized handwritten data that adheres to the style of the provided reference samples while maintaining the content of the template. We utilize diffusion stochastic differential equations, which have demonstrated strong capabilities in modeling data distributions to characterize both of these processes. Our contributions can be summarized as follows:

- For the first time, we accomplish the generation of handwritten Chinese text images at the line levels using two-stage diffusion.

- We propose CharPos-Diff, a layout method based on generative diffusion models designed for arranging text lines. This method generates a **standard text line template** for the input content, ensuring that it aligns with the positional layout characteristics of the provided style reference sample.

- We propose Imitating-Diff, which can effectively integrate content and style features, facilitating the transition from the standard **text line templates** to target style samples while preserving the structural integrity of **line-level** glyphs and ensuring the writing style aligns with the desired target style.

- We introduce a loss function for better aligning style and content, which enhances the quality of generated handwriting images through fine-tuning.

## 2  RELATED WORK

Handwritten data can be stored in the form of online trajectory sequences or offline images. The advantage of online handwritten data lies in its ability to maintain dynamic trajectory information of the drawing process, while the advantage of offline handwritten data lies in its more natural representation of visual features such as stroke width and ink intensity.

**Online Handwriting Generation.** Most of online handwriting generation methods are typically based on autoregressive neural network (Graves, 2013), utilizing a mixture of Gaussian distributions to model the movement of pen. SketchRNN (Ha & Eck, 2018), which based on Variational Autoencoders (VAE) (Kingma & Welling, 2014) adopts RNN to conduct the task of unconditional sequence data generation. Subsequently, a series of works (Aksan et al., 2018; Kotani et al., 2020; Tolosana et al., 2021; Tang et al., 2019; Zhao et al., 2020; Tang & Lian, 2021) focuses on conditional generation, aiming to produce handwriting that encompasses specific content or diverse writing styles. In addition, there are studies that apply diffusion models to online sequential data (Luhman & Luhman, 2020; Das et al., 2023; Ren et al., 2023), achieving notable results.

**Offline Handwriting generation.** Offline handwriting generation methods (Alonso et al., 2019; Xie et al., 2021; Bhunia et al., 2021; Gan & Wang, 2021; Kong et al., 2022; Liu et al., 2022) were primarily based on Generative Adversarial Networks (GANs) in the early days, leveraging adversarial training to create realistic handwriting samples. Recently, with the development of diffusion models, an increasing number of studies have begun to utilize these models to generate offline font data Yang et al. (2024). Diffusion models offer a novel framework that refines random noise into structured handwriting, allowing for more nuanced control over style and content. State-of-the-art methods can now extract style information from a significantly limited number of reference samples Yao et al. (2024); Dai et al. (2024).

**Score-Based Diffusion Model.** Score-based diffusion models systematically perturb data via a forward diffusion process, subsequently reversing this process to recover the original data (Sohl-Dickstein et al., 2015; Song & Ermon, 2019; Ho et al., 2021). Due to their capability to balance fidelity and diversity, along with advancements in accelerated sampling (Song et al., 2020; Bao et al., 2021; Lu et al., 2022) and conditional generation (Dhariwal & Nichol, 2021; Rombach et al., 2021), these models have been extensively adopted across diverse data modalities. They currently occupy a preeminent position in the realm of image generation tasks (Ho et al., 2021; Dhariwal & Nichol, 2021). We also adopt diffusion models as our generative framework.

## 3  METHODS

### 3.1  PRELIMINARY

**Problem Statement.** Our goal is to synthesize handwritten text images given an arbitrary length Chinese string $\mathcal{A}$ and a single style sample $I_s$ from a writer $w_s$. The generated handwritten text images should reflect the distinctive handwriting style of $w_s$ while accurately representing the content of the Chinese string $\mathcal{A}$. This entails ensuring that the text line's content closely adheres to the specified string, with all individual character structures precisely depicted. Additionally, the spatial arrangement between characters must be fluid, preserving the natural rhythm of handwritten text.

Existing related methods *are limited to generating single Chinese characters* and cannot handle more complex text content. For one thing, these methods do not consider the positional relationships among characters. By generating individual characters followed by simple concatenation will result in a lack of coherence in the font structure within the text line. For another, if directly applied to

generate handwritten text lines combined with templates generated by our method, these methods typically lead to content structure errors in the generated text lines.

**Diffusion Formulation.** Following (Song et al., 2021), we formulate the diffusion (forward) process and denoising (reverse) process as stochastic differential equations (SDEs). The forward SDE transforms the true data vectors $x_0$ into Gaussian noise:

$$dx_t = -\frac{1}{2}\beta_t x_t dt + \sqrt{\beta_t} dw_t, \quad t \in [0, T], \tag{1}$$

where $w_t$ represents standard Brownian motion, $t \in [0, T]$ with $T$ as a finite time horizon, and $\beta_t$ is a non-negative noise schedule function. The solution to this equation is given by:

$$x_t = e^{-\frac{1}{2}\int_0^t \beta_s ds} x_0 + \int_0^t \sqrt{\beta_s} e^{-\frac{1}{2}\int_0^t \beta_u du} dw_s. \tag{2}$$

By utilizing properties of Ito's integral, the conditional distribution of $x_t$ given $x_0$ is Gaussian:

$$p(x_t|x_0) \sim \mathcal{N}(\rho(x_0, t), \lambda_t), \tag{3}$$

where $\rho(x_0, t) = e^{-\frac{1}{2}\int_0^t \beta_s ds} x_0$ and $\lambda_t = I - e^{-\int_0^t \beta_s ds}$. The reverse SDE transforms the Gaussian noise back into the data $x_0$ using the following process:

$$dx_t = -\left(\frac{1}{2}x_t + \nabla \log p_t(x_t)\right)\beta_t dt + \sqrt{\beta_t} d\tilde{w}_t, \quad t \in [0, T], \tag{4}$$

where $\tilde{w}$ denotes the reverse-time Brownian motion. Additionally, we can represent the reverse process as an ordinary differential equation (ODE):

$$dx_t = -\frac{1}{2}\left(x_t + \nabla \log p_t(x_t)\right)\beta_t dt, \quad t \in [0, T]. \tag{5}$$

We train the network as denoiser $\mathcal{E}_\theta$ to predict the original clean data from noisy data:

$$\hat{x}_0 = \mathcal{E}_\theta(x_t, t, c),$$

where $c$ represents the additional conditions. We use the predicted $x_t = 0$ to estimate the score $\nabla \log p_t(x_t)$ using $\lambda_t^{-1}(\rho(\hat{x}_0, t) - x_t)$, allowing us to sample data $x_0$ by starting from Gaussian noise $x_T \sim \mathcal{N}(0, 1)$ and numerically solving the SDE in Equation equation 4 or the ODE in Equation equation 5. The loss function used to train the diffusion model is expressed as follows:

$$\ell_{\text{diff}} = \mathbb{E}_{x_0, t}[\|\hat{x}_0 - x_0\|_2^2]. \tag{6}$$

### 3.2 OVERVIEW

As shown in Figure 1, we first synthesize standard templates for text lines and then transfer these templates to specified writing style images.

**Text Line Content Template.** The standard text line template consists of *the bounding box of each character* and *the standard character template*. More concretely, the bounding box decides the location and the size of a character in the whole text line image. The character category decides which font image (i.e. SimHei font) to use for a character template. All bounding box coordinates are changed to height, width, vertical distance of each character and the distance between the centers of adjacent characters with normalization. We fix the number of bounding boxes of style images and content images to be 32, which is adequate based on the writing habits of the majority of individuals.

Based on the bounding boxes and single character templates, we seamlessly make text line template images by integrating the SimHei image of each character into a blank image with size of 96x2048 pixels. All text line images are resized to 96 pixels in height, preserving their aspect ratio and then padding to 2048 pixels in width.

#### 3.2.1 CHARPOS-DIFFUSION

As mentioned before, to construct reasonable standard character templates for text of arbitrary length, it is crucial to generate a coherent and reasonable layout for the text lines (bounding boxes). Specifically, considering that different characters exhibit distinct shapes, we learn a unique encoding for each character class i: $\mathbf{Enc}_i \in \mathbb{R}^d$, where $d$ is the dimension of encoding vector. The generation

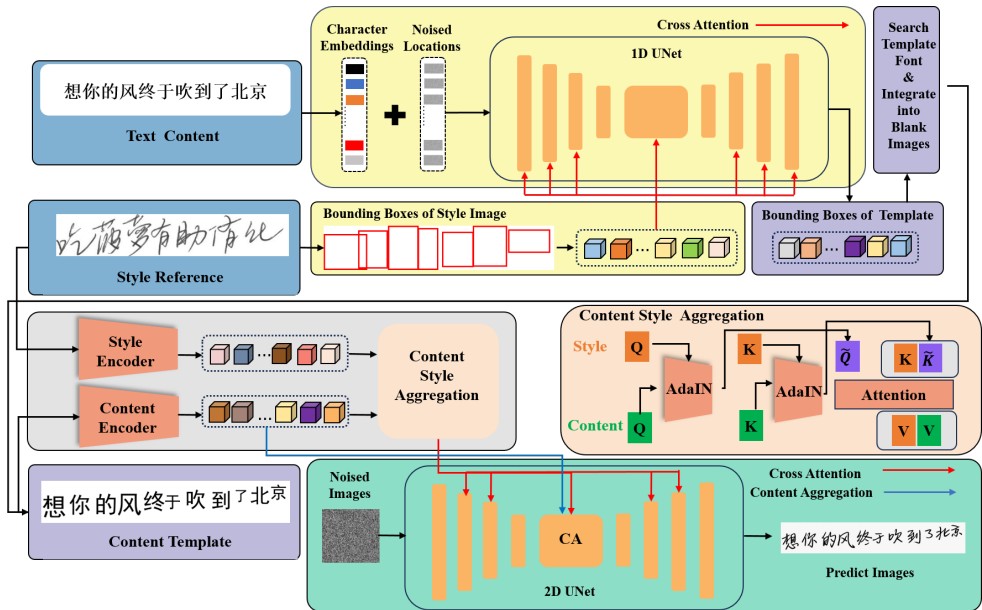

Figure 1: The overview framework of our methods, which consists of the Char-Pos Diffusion and the Imitating-Diffusion.

process is modeled through a diffusion framework, employing a one-dimensional U-Net network as the denoising model $\mathcal{E}_p$. Additionally, recognizing that *different authors exhibit distinct layout styles in their handwriting*, we incorporate the layout of style reference samples as a condition. After passing through a linear mapping layer, this reference layout information $Layout_{ref}$ is input into the denoiser via cross-attention.

**Training Objective.** As stated in Section 3.1, we model the reconstruction loss as the average distance between the predicted clean data and the actual data. Recalling equation 6, here $\boldsymbol{x}_0$ represents the ground truth bounding boxes and $\hat{\boldsymbol{x}}_0$ represents the predicted ones:

$$\ell_{\text{diff-pos}} = \mathbb{E}_{\boldsymbol{x}_0,t}[\|\hat{\boldsymbol{x}}_0 - \boldsymbol{x}_0\|_2^2], \quad \hat{\boldsymbol{x}}_0 = \mathcal{E}_p(\mathbf{Enc}_i, Layout_{ref}) \tag{7}$$

### 3.2.2 IMITATING-DIFFUSION

**Content Aggregation Module.** Following (Yang et al., 2024), we adopt the Content Aggregation (CA) like them. We take CG-GAN (Kong et al., 2022) as our content encoder and style encoder. The differences lies in that they inject multi scale content features into the UNet model while we only inject the content features with the highest dimension into the UNet mid module, not including the downsampling blocks. We find that this content injection method is so powerful that the injection of multi-scale content features is harmful for model's learning good style representations, especially for handwritten characters with more styles than printed font. The content features are firstly concatenated with noised inputs. Then we apply a channel attention, including an average pooling, two convolutions with the kernel size of 1, and an activation function. The channel attention output vector is used to weight the concatenated features through channel-wise multiplication. After a residual addition, we scale the channel number of the weighted vector to the output channels.

**Content Style Aggregation Module.** We propose a content style aggregation module (CSA) to replace the direct cross attention for style conditions. After content encoder and style encoder, we get the high level content features and style features. Like transformer attention module, we acquire Q, K, V for content and style features. AdaIN is then adopted to shift the content Q distribution to the style Q distribution. The same method is operated to K. Then we concatenate the shifted content K and the style K while we concatenate the origin content V and the style V. In the end, we do scaled attention operation with the shifed Q, concatenated K and the concatenated V, reaching the final aggregated features.

**Training Objective.** Recalling equation 6, here $\boldsymbol{x}_0$ represents the origin image and $\hat{\boldsymbol{x}}_0$ represents the predicted ones. Inspired by (Yao et al., 2024), we *place greater emphasis on the contour information of strokes rather than on blank areas*. Therefore, we incorporate weights based on Harris corner detection into the existing loss function, which can be expressed as:

$$\ell_{diff'} = \mathbb{E}_{x_0,t}[\omega(\hat{\boldsymbol{x}}_0) \circ [\|(\boldsymbol{x}_0 - \hat{\boldsymbol{x}}_0)\|^2]],$$
$$\omega(\hat{\boldsymbol{x}}_0) = \lambda_{corner} \cdot Harris(\hat{\boldsymbol{x}}_0) + \lambda_{all} \cdot \hat{\boldsymbol{x}}_0. \tag{8}$$

We set $\lambda_{corner} = 0.9$ and $\lambda_{all} = 0.1$ in our experiments.

Besides, we employ a perceptual loss function to enhance the quality of the generated images. This perceptual loss, denoted as $\ell_{cp}$, is defined as follows:

$$\ell_{cp} = \sum_{l=1}^{L} \|VGG_l(\boldsymbol{x}_0) - VGG_l(\boldsymbol{x}_{target})\|, \tag{9}$$

where $VGG_l(\cdot)$ represents the feature maps obtained from the $l$-th layer of the VGG network, and $L$ is the total number of chosen layers. The total loss function is expressed as:

$$\ell_{total} = \ell_{diff'} + 0.01 \cdot \ell_{cp}. \tag{10}$$

**Fine-Tuning Function.** To facilitate the alignment of style and content, we *duplicate the style and content encoders* after a certain number of training epochs and *fix their weights*. This allows us to extract the style and content features of both the reference samples and the generated samples. We denote the distance vector of the style and content features for the reference sample as $\mathcal{D}_{cs1}$ and for the generated sample as $\mathcal{D}_{cs2}$. The alignment loss function is defined as:

$$\ell_{align} = 1 - \frac{\mathcal{D}_{cs1}\mathcal{D}_{cs2}}{\|\mathcal{D}_{cs1}\|\|\mathcal{D}_{cs2}\|}. \tag{11}$$

During the fine-tuning phase, the loss is defined as: $\ell_{total'} = \ell_{total} + 0.1 \cdot \ell_{align}$.

## 4 EXPERIMENTS

Our method can generate handwritten Chinese text line images of arbitrary length, where individual characters can be considered as text lines of length one. Therefore, on one hand, we validate the effectiveness of our approach to generate text lines. On the other hand, we also compare individual characters with the current state-of-the-art methods.

### 4.1 DATASET.

**Single Character Dataset.** We conduct single-character generation experiments on the ICDAR-2013 competition database (Yin et al., 2013), which consists of 60 writers and 3755 different character for each writer. We randomly select 80% of the dataset as the training set and the remaining 20% as the test set. We adopt the Unifont for content images. In the experiments, we resize all single-character images to $64 \times 64$.

**Full Textline Dataset.** To evaluate the ability of generating full handwritten text lines. We use the Chinese offline dataset CASIA-HWDB2.0-2.2Liu et al. (2011), which consists of 52230 Chinese textlines from 1019 different writers. We randomly select 41781 textlines from 816 writers for training and the remaining 10449 textlines from 203 writers for testing.

### 4.2 IMPLEMENT DETAILS

In all the experiments, we use *only one style reference sample* to perform one-shot experiments (one font for character generation and one text line for layout and text line generation). We train all diffusion models with classifier-free guidance strategy under the guidance scale of 0.1 on four A6000 GPUs. Specifically, we train layout diffusion model for 60000 steps (batch size of 1024) , text line diffusion model for 200000 steps with batch size of 4 and single-character diffusion model for 66000 steps with batch size of 168. AdamW optimizer with $\beta_1 = 0.9$ and $\beta_2 = 0.999$ is used. The learning rate is set as $1e - 4$ with linear schedule. During inference, DPM-solver++(Lu et al., 2022) is used to speed up sampling with 25 steps. We fine-tune our model for 30000 iterations with the proposed content-style alignment loss.

### 4.3 EVALUATIONS

#### 4.3.1 SINGLE CHARACTER GENERATION

Table 1: Compared with SOTA Method.

|  | Fid | Style Score | Content Score | L1 | LPIPS | SSIM |
|---|---|---|---|---|---|---|
| One-DM | 32.20 | 0.5620 | 0.8218 | 0.1241 | 0.1291 | 0.2776 |
| CSA + finetune | 32.62 | 0.4650 | 0.9880 | 0.1458 | 0.1340 | 0.26198 |

Due to the lack of baseline for Chinese handwritten textline generation task, we conduct comparison and ablation experiments in single-character generation tasks. As shown in Table 1, our Content Style Aggregation (CSA) effectively improves the generation quality compared to the method that only style features are injected into model with cross attention. From Figure 2, we observe that models with CSA can generate more stylized character structure than those without CSA. After finetuned with content-style alignment loss, models with CSA can generate characters with better ink color and stroke thickness. It strongly proves the effectiveness of our proposed loss.

For comparison experimenet, we choose One-DM model with the same experiment settings. Though results of One-DM's generation quality show slightly better than our model in style imitating, One-DM is **struggling to generate characters with correct and recognizable structures than our model**. It seems that One-DM focuses more on ink color and stroke thickness. The style model we trained on the authentic test dataset appears to primarily distinguish character writers based on ink color and stroke thickness. In contrast, people tend to identify the author by analyzing the character structures.

Table 2: Ablation of CSA Module.

|  | Fid | Style Score | Content Score | L1 | LPIPS | SSIM |
|---|---|---|---|---|---|---|
| No CSA | 48.63 | 0.0660 | 0.9992 | 0.1786 | 0.1535 | 0.2375 |
| CSA | 45.15 | 0.2251 | 0.9921 | 0.1610 | 0.1457 | 0.2552 |
| CSA + finetune | **32.62** | **0.4612** | **0.9880** | **0.1458** | **0.1340** | **0.2619** |

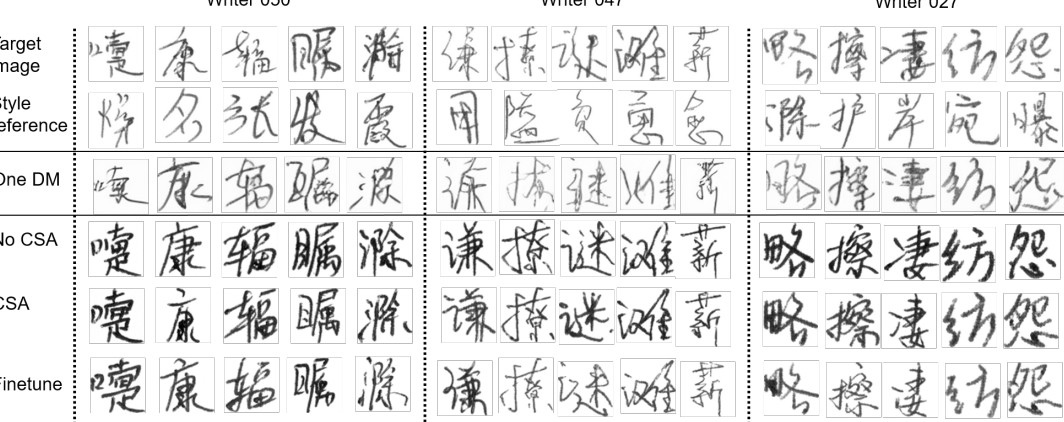

Figure 2: The generated samples by different methods.

#### 4.3.2 TEXT LINE GENERATION

**Evaluation of Layout.** We compare our LayoutDiffuser with other layout generation models. The results are shown in Table 3. The LayoutTransformer model is capable of generating stylized layouts by utilizing fixed-length style reference bounding boxes in an autoregressive method. LayoutLSTM follows the same approach, with the primary difference being the substitution of the transformer blocks with LSTM layers. Through this in-context approach, the model can learn the style characteristics (e.g. textline slant) of the reference sample to a certain degree. However, Figure 3 shows that both LayoutTransformer and LayoutLSTM underperform compared to the LayoutDiffuser model in

terms of generating accurate character spacing and character size. We hypothesize that this underperformance may be attributed to exposure bias inherent in autoregressive models during the inference process, leading to cumulative errors in text line generation. In contrast, diffusion models are less susceptible to this issue.

Table 3: The comparison of different layout generative methods.

| | Height Loss | Width Loss | Vertical Loss | Horizontal Loss |
|---|---|---|---|---|
| LayoutLSTM | 0.0877 | 0.0780 | 0.0805 | 0.0946 |
| LayoutFormer | 0.0842 | 0.0743 | 0.0816 | 0.0911 |
| LayoutDiffuser | **0.0179** | **0.0157** | **0.0145** | **0.0269** |

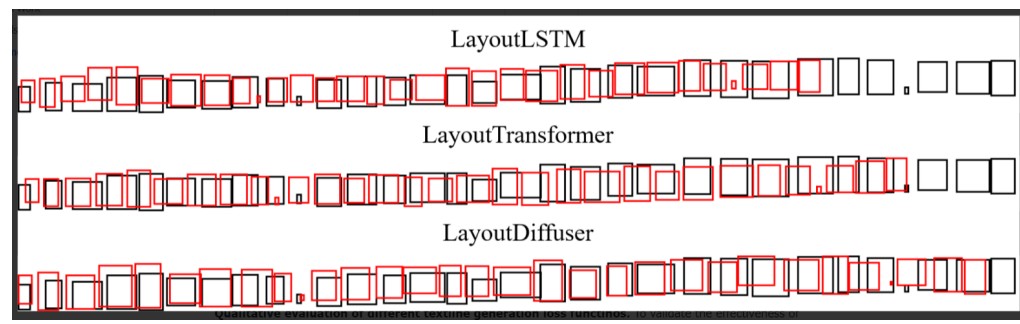

Figure 3: Visualization of layout generated by different methods, where black represents the ground truth.

**Evaluation of Full Line.** As shown in the Figure 4, our method can effectively generate complete handwritten text line images with accurate structure and diverse styles.

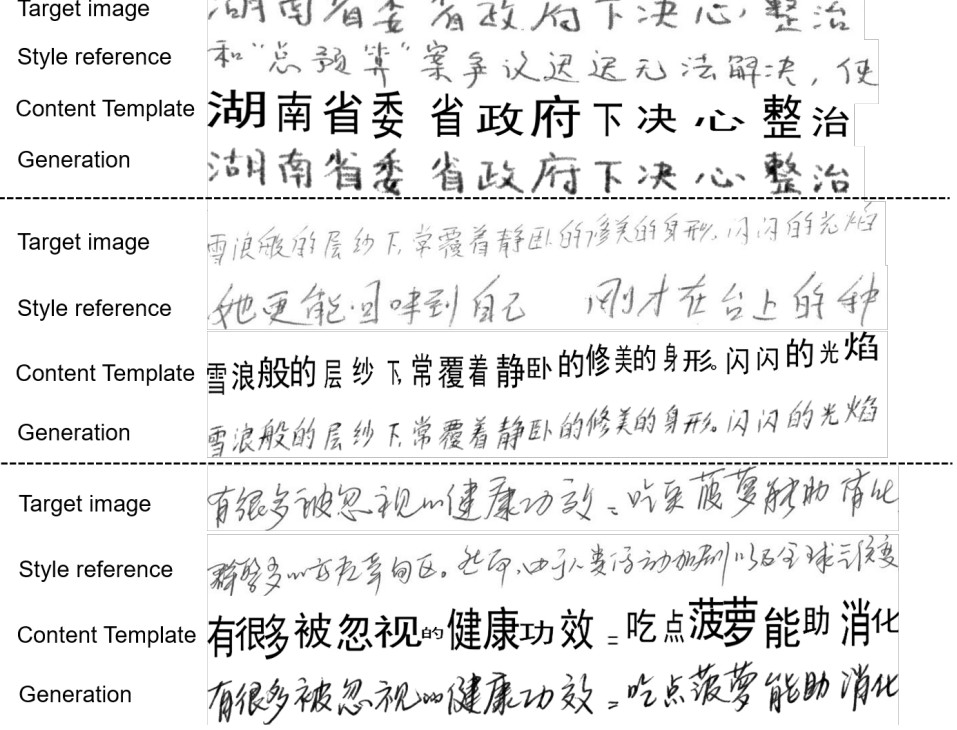

Figure 4: Visualization of the generated handwritten text lines.

**Ablation of Loss Function.**

To validate the effectiveness of our loss in stylized Chinese handwritten text line generation, we conduct visual experiments. As shown in Figure 5, we can observe that all the loss functions enable the model to generate characters in the desired positions, provided that content information is appropriately integrated. For the L2 loss, the model fails to focus on the foreground of the text line, namely the character components, resulting in the generation of indistinct character structures and noisy backgrounds. To improve generation quality, an effective and direct approach is to utilize the bounding box information of each character, allowing the model to focus exclusively on the L2 loss function within the confines of the bounding boxes. This approach indeed enables the model to generate text lines with a relatively clean background; however, the model's ability to imitate the style of the text lines remains insufficient. This is evidenced by the overly thick strokes, lack of ligatures, and the presence of considerable noise around the characters. Considering the Harris corners of the target text line image, the model trained with our loss function is better equipped to capture the fine-grained style information of characters, more precisely focusing on the salient components within the text line image.

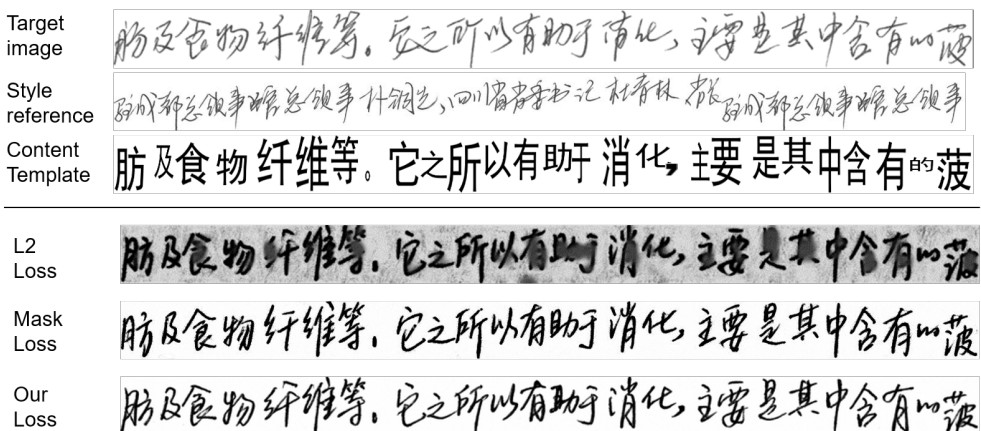

Figure 5: Visualization of the generated handwritten text lines using different loss function.

## 5 CONCLUSION AND FUTURE WORK

In this paper, we have presented a novel approach for generating offline Chinese handwritten text lines. By modeling handwritten text generation as a style transfer problem, we developed a two-stage method to handle this unexplored task. The first stage generates character positions through Character-Position-Diffusion, creating structured text line templates. The second stage employs an Imitating-Diff model for font style transfer, producing coherent and stylistically accurate handwritten text lines images. However, as shown in Figure 2, when imitating fonts with lighter ink colors, our generated sample is slightly darker. *Actually, the focus of the model on content versus style is a matter of balance. This issue highlights the need for further refinement in the content and style control modules* to address such discrepancies. Additionally, our approach theoretically possesses significant generalizability, making it worthwhile to explore its application to other forms of handwritten data, including various languages and mathematical formulas. We consider these areas as potential directions for future research.

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
