# OpenReview forum: "Write More at Once: Stylized Chinese Handwriting Generation via Two-stage Diffusion"
_ICLR.cc/2025/Conference — ICLR 2025 Conference Withdrawn Submission_

### Official Review · Reviewer_MecL · 2024-11-01

**Soundness:** 3
**Presentation:** 3
**Contribution:** 2
**Rating:** 6
**Confidence:** 4

**Summary:**

The paper proposes an approach for the style-conditional generation of images of lines of Chinese handwritten text. The proposed model is a combination of two diffusion-based models. First model takes a sample of the handwriting as a source of style, and the text that needs to be generated, and produces a template which places (still typeset) characters on a line, adjusting their size and positions to match the style source image. Second model takes the template, the style image, and the desired text content, and generates the desired output, ensuring that the placement and sizing of the characters follows the template generated by the first model, the style of the letters follows the style in the provided style sample, and the content matches the input text string. Both models are U-Net based.  The performance is measured using several standard metrics such a sFID score of the generated images, and outperforms several previous approaches. Additionally, the quality of template generation only is compared to some other approaches in the literature, LSTM- and Transformer-based. Finally, authors propose a content-style aggregation block as a part of their architecture, which is replacing the standard attention mechanism and seems to perform favorably compared to it.

**Strengths:**

Originality: The paper presents an approach to image generation of Chinese handwritten lines. The task itself and the separation into layout and content generation is not novel, but the diffusion approach to this problem is somewhat novel.
Quality & Clarity: The writing is mostly clear. The ablation study is a strong part of the paper, investigating the quality of the content generation on the individual symbols dataset, the quality of layout generation independently from the content generation, and the contribution of the proposed content style aggregation model.

**Weaknesses:**

I believe that the main weakness of the paper is the combination of fairly narrow domain / significance and the absence of the open-source model/code release:

* The proposed approach is particularly suitable for languages that don't have cursive writing but doesn't seem to generalize beyond that.
* The proposed layout generation approach could be suitable for other types of problems, such as generation of pages of handwriting or multi-line handwriting, but is evaluated only on the narrow domain.
* Given the narrow focus of the paper and a fairly complex model consisting of two diffusion models and a particular attention mechanism, reproducibility of this approach is fairly difficult, thus hindering further progress based on this work.

I believe that the paper could be strengthened either by releasing the code or the model, or by showcasing that the proposed approach could generalize beyond the domain highlighted in the paper (ex. different scripts or different types of images)

**Questions:**

Section 4.1 title "Dataset." --> "Dataset"
Section 4.2 title "Implement details" --> "Implementation details"

---

### Official Review · Reviewer_9pau · 2024-11-01

**Soundness:** 2
**Presentation:** 2
**Contribution:** 3
**Rating:** 5
**Confidence:** 5

**Summary:**

This paper introduces a two-stage diffusion model for generating stylized Chinese handwritten text lines. Unlike previous methods limited to single font outputs, this approach tackles sentence-level generation by treating it as a image style transfer problem. The first stage, CharPos-Diff, generates the character positions and creates a text line template using standard font images. The second stage, Imitating-Diff, then transforms this template into a handwritten style using a diffusion model, incorporating style information from a reference sample. The model incorporates a novel loss function that emphasizes stroke contours and uses a content-style alignment technique for improved results. Experiments demonstrate the method's effectiveness in generating structurally accurate and stylistically consistent handwritten text lines.

**Strengths:**

1.	It is good to see the work about generating handwritten Chinese text lines. Although there have been some studies on generating handwritten English text lines, the more challenging task of generating handwritten Chinese text lines is still a relatively under-explored area.

2.	The core idea of this work (first generate a template image and then transfer the template into a stylized image.) is well-motivated and novel.

3.	The proposed two-step diffusion-based method is technically sound and validated by experiments.

**Weaknesses:**

1. Overall, the experimental section can be refined with more explanation. For example, （1） there have been several existing works on single handwritten Chinese character generation, but this paper only compares one method; moreover, from the results in Tab. 1, the performance of the proposed method is only similar to that of One-DM. （2） there is no quantitative evaluation experiments for the generated text line images. (3) Tab. 2 is not referred in the paper. (4) Performance metrics used in Tab. 1 are not explained.

2. The introduction to IMITATING-DIFFUSION is a little simplistic, making it not easy to fully understand. It only explains how each module is implemented, but does not describe how the modules interact with each other. In addition, most of the design seems to be based on existing methods.

**Questions:**

In table 2, as the CSA module can extract both content and style information, what would happen to the method's performance if the CA module is removed and only the CSA module is retained?

---

### Official Review · Reviewer_daHJ · 2024-11-03

**Soundness:** 1
**Presentation:** 1
**Contribution:** 2
**Rating:** 3
**Confidence:** 5

**Summary:**

This paper proposes a two-stage diffusion method to synthesize Chinese handwritten text line, conditioned on text content and style reference image. The key idea is to first generate the layout that consists of the text content and character positions, then combine the layout and character style information to synthesize the desired text line. Some experiments evaluate the proposed method.

**Strengths:**

1)	It is interesting to break down the text line generation into two independent processes.
2)	The proposed LayoutDiffuser component generates accurate layout.

**Weaknesses:**

1)	The pipeline of the proposed method is unclear. It is recommended to provide an overall introduction of the whole method in Section 3.2.
2)	I am confused about how to obtain the bounding boxes of style image in the testing phase.
3)	The proposed LayoutDiffuser lacks a definition and introduction in the method section.
4)	The effectiveness of the proposed method is questionable, as Table 1 shows it lags behind the SOTA method One-DM across five metrics, such as FID and style score.
5)	In Figure 2 and the last row of Figure 4, the generated samples differ significantly from the Target Image in terms of ink color, and stroke connections, raising doubts about whether the proposed method can accurately mimic the handwriting style.
6)	Quantitative ablation results are recommended to provide.
7)	This paper does not include a user study and an analysis of failure cases.

**Questions:**

My major concerns are the unclear method description and weak experiment designs. More details are provided in Weaknesses.

---

### Official Review · Reviewer_NPee · 2024-11-03

**Soundness:** 2
**Presentation:** 2
**Contribution:** 2
**Rating:** 5
**Confidence:** 4

**Summary:**

The paper presents a method for generating stylized Chinese handwriting at the sentence level using a two-stage diffusion model. The model addresses limitations in previous methods that focused on generating single characters or words. The authors introduce two primary components:
    1. CharPos-Diffusion for generating character positions in a text line.
    2. Imitating-Diffusion for transferring the layout into a specified handwriting style.
This approach  models text-line generation as a style transfer problem and enables the generation of coherent sentence-level handwriting with personalized style features.
 The CharPos-Diffusion stage generates a layout of character positions, creating a structured template for the text line based on reference samples. Whereas the Imitating-Diffusion stage uses these templates to transfer style information from a reference, combining content and style features effectively.

**Strengths:**

1. The paper identifies an essential gap in  “Chinese handwritten text generation” research: the need to go beyond isolated character generation. By focusing on full text lines, this method supports applications that require sentence-level coherence.

2. The "CharPos-Diffusion" component introduces a  layout loss that maintains the "positional relationships" between characters, ensuring consistent spacing and alignment.
3. The "imitating-Diffusion" stage incorporates Harris corner detection into its loss function to emphasize stroke contours. This inclusion enhances style fidelity by focusing on the stroke details.

**Weaknesses:**

1. The two-stage diffusion process, combined with the need for high-dimensional style and content features, leads to **high computational demands**. This complexity could limit the model’s scalability, particularly for real-time applications.

2. This approach is only suitable to generate text lines from Scripts with distinct separate characters with a clear bounding box, this is not a generic approach which could be applied to generate cursive handwritten text of Latin script.

3. Time complexity analysis of the entire method is missing,

4. Ablation of loss function - involves only qualitative method.

**Questions:**

1. Generating samples from a score-based diffusion model requires a large number of steps.  The authors have mentioned  that they have used  a reduced-step solver like DPM-Solver++ to speed up the sampling. But most likely the consequences of using it is  reduced image quality or stability. It is unclear from the current draft what measures has been taken to address that issue.

2. Column Captions/heading title in Table 1 and Table 2 – What is meant by style score and content score? Are those referring to Writer Identification accuracy and character recognition accuracy?

3. Dataset train /test splitting: In  page 5, line 254 and 258 the authors have mentioned that they have randomly selected training and testing data. How could the results be reproducible in such a framework? This is important for fair comparison  and bench-marking purpose.

4. Can the model  generate text with an unseen style ?

---

### Note · Authors · 2024-11-14

I have read and agree with the venue's withdrawal policy on behalf of myself and my co-authors.